# Towards an Effective Service Allocation in Fog Computing

**DOI:** 10.3390/s23177327

**Published:** 2023-08-22

**Authors:** Rayan A. Alsemmeari, Mohamed Yehia Dahab, Badraddin Alturki, Abdulaziz A. Alsulami, Raed Alsini

**Affiliations:** 1Department of Information Technology, Faculty of Computing and Information Technology, King Abdulaziz University, Jeddah 21589, Saudi Arabia; ralsemmeari@kau.edu.sa (R.A.A.); baalturki@kau.edu.sa (B.A.); 2Department of Computer Science, Faculty of Computing and Information Technology, King Abdulaziz University, Jeddah 21589, Saudi Arabia; mdahab@kau.edu.sa; 3Department of Information Systems, Faculty of Computing and Information Technology, King Abdulaziz University, Jeddah 21589, Saudi Arabia; ralsinie@kau.edu.sa

**Keywords:** IoT, IoMT, fog computing, service allocation, optimization, cloud computing

## Abstract

The Internet of Things (IoT) generates a large volume of data whenever devices are interconnected and exchange data across a network. Consequently, a variety of services with diverse needs arises, including capacity requirements, data quality, and latency demands. These services operate on fog computing devices, which are limited in power and bandwidth compared to the cloud. The primary challenge lies in determining the optimal location for service implementation: in the fog, in the cloud, or in a hybrid setup. This paper introduces an efficient allocation technique that moves processing closer to the network’s fog side. It explores the optimal allocation of devices and services while maintaining resource utilization within an IoT architecture. The paper also examines the significance of allocating services to devices and optimizing resource utilization in fog computing. In IoT scenarios, where a wide range of services and devices coexist, it becomes crucial to effectively assign services to devices. We propose priority-based service allocation (PSA) and sort-based service allocation (SSA) techniques, which are employed to determine the optimal order for the utilizing devices to perform different services. Experimental results demonstrate that our proposed technique reduces data communication over the network by 88%, which is achieved by allocating most services locally in the fog. We increased the distribution of services to fog devices by 96%, while simultaneously minimizing the wastage of fog resources.

## 1. Introduction

IoT devices generate a large amount of data as they are interconnected [1]. Most current proposals focus on centralized or cloud architecture [2]. The goal of a centralized architecture is to process data in one place of decision. Consequently, a significant amount of data must be uploaded to the cloud. Heavy data transmission over the network is one of the challenges introduced by this architecture [3]. This suggests that an alternative design is necessary to address this challenge. Since the IoT architecture connects several devices with varied levels of computing, storage capacity, battery life, and Internet access, device constraint awareness is a crucial part of its design.

Also, a variety of services will be available, each with different set of expectations, such as those for capability, quality of data, and latency. These services operate on fog computing devices, which are limited in terms of power when compared to cloud resources [4], and they require bandwidth. This implies that fog devices and services are closely connected. The main challenge is to allocate the services primarily to the fog whenever possible, and then to the cloud when the fog is not capable of handling services while considering the overall efficiency within a specific IoT architecture.

Furthermore, resource management at the fog layer [5] is critical for evaluating the advantages of fog computing. Developing an effective fog infrastructure presents several issues. One instance where resolving the following issues becomes essential is with RAM. The execution of services within a distributed architecture becomes more challenging as the size and complexity of the IoT system increase, necessitating a method to allocate services to the node(s), which results in discovering the ideal allocation strategy.

Processing all incoming raw data in the cloud has a detrimental impact on various elements, including higher network congestion, latency, the time it takes to return actions to a user, energy usage, and privacy [6]. As the Internet of Things expands, there is a need to address these challenges. Fog devices are restricted devices because of their limited computing power when compared to the cloud. As a result, huge workloads cannot be processed on fog devices. In addition, determining the amount of computing load that may be allocated to a fog device is challenging. Furthermore, distributing services among fog devices is difficult since the large number of services in the IoT might demand a lot of computing power [7]. Effectively allocating services to fog devices with limited resources in an IoMT system presents a challenge. This necessitates the allocation of services based on priority to both fog and cloud devices while considering resource constraints and optimizing for efficient processing. As a result, we must understand the devices’ capabilities and services’ demands. Next, we must optimize the process of service allocation to the devices while maintaining optimal resource utilization.

This paper’s overarching goal is to provide an effective allocation technique for processing data with reduced bandwidth utilization, faster response time, optimized resource usage, and identifying an optimal approach for processing data on a large scale. One of the objectives is to evaluate and test the proposed technique using a simulation. We propose an efficient allocation strategy that brings processing closer to the fog side of the network. Moreover, we investigate which devices and services may be best allocated while preserving resource use in the IoT architecture. In addition, we offer a service allocation technique for allocating services to devices depending on their capabilities. Our main contributions include the following:Service allocation techniques are significant because providing services to devices in the IoT is a difficult process due to the many types of devices and their capabilities. As a result, we propose priority-based service allocation (PSA) and sort-based service allocation (SSA) techniques, which utilize a list of every fog device connected to the network. This method makes it possible to use fog devices in the best possible sequence to conduct a wide range of services. As a starting point, we use packing problems as a baseline to help solve allocation issues in the IoT environment.We examine the importance of allocating services to devices and optimizing resource use in fog computing to enhance service quality while meeting the optimal resource usage demands of IoMT. As there will be a large variety of services and devices in the IoT settings, it is vital to allocate the services to the devices and effectively optimize resource consumption.We evaluate the PSA and SSA techniques using a synthetic dataset that mimics the IoT services and devices. We perform a tradeoff analysis to illustrate the effectiveness of the service allocation approach. The results reveal that the data communication over the network decreased by 96%, as most services are allocated in the fog. Additionally, latency is reduced by approximately 88%.

The remainder of this paper is organized as follows: Section 2 presents related works in the field of service allocation; Section 3 describes the research problem and provides a motivational scenario; Section 4 provides the methodology, including the algorithm and the architecture; Section 5 presents the experimental setup and reveals the details of the experiments; Section 6 shows the results that were obtained from experiments and provides a description of the results; followed by discussion and evaluation; finally, Section 8 presents our conclusions and recommendations for future research.

## 2. Related Works

Fog computing has become an increasingly popular topic of research in recent years, as it offers a number of benefits for various industries [8]. One of the main challenges in fog computing is data distribution and allocation. This literature review aims to explore the current state of research on service allocation in fog computing and highlight key references in the field.

Analyzing data closer to the fog leads to reduced latency, and increased efficiency, as well as improved security and privacy [9]. Fog computing also enables the deployment of computing resources closer to the data source, reducing the need for data transmission over long distances. This can lead to improved performance and lower energy consumption.

Regarding service allocation, research has focused on the use of optimization techniques to allocate resources efficiently in fog computing environments. Optimization techniques such as game theory, packing, linear programming, and scheduling can also be used to model and solve service allocation problems in fog computing environments.

In the context of resource allocation in edge and fog computing, we reviewed research publications that focus on fog systems. The underlying infrastructure is assumed in these studies to be cloud–fog [10,11,12,13,14,15,16,17,18,19,20,21,22,23,24,25,26]. These options fall under the system elements aspect and have a significant impact on the researchers’ optimization goals. When considering cloud and fog computing, many academics believe that the workload is initially stored in the cloud, and the edge system must decide where to duplicate and how to distribute the user load among them [11]. As a result, they offer a framework for pushing applications that require lots of resources to the fog and reducing average data communication in the edge network across access points by duplicating cloud services on some of the edge servers. Workload distribution over systems that are heterogeneous has to consider the availability of various resources [27]. When spreading the workload between fog and cloud, the objective is to reduce energy consumption in order to meet service latency needs [28]. When a specific research project does not assume the use of a central cloud but instead addresses multi-fog situations, the issue may arise from the combined optimization of job distribution, virtual machine placement, and resource allocation [12]. The authors in [14] attempt to reduce the load on users by determining user association, joint service placement, and joint allocation.

However, most of the publications have the same optimization objective(s), namely, service completion latency [12,15,16,20,21,22,23]. Numerous research endeavors have been undertaken to address the trade-off between energy consumption and delay in data transmission [27,28]. In addition to providing prompt service completion to users, researchers aim to accommodate numerous users with the edge fog [19,22]. Cost minimization includes several aspects, such as resource usage, quality of service, and its associated revenue. The authors of [29] calculated the total cost of deployment by considering the wireless communication cost and the function placement computation cost, and the authors of [13,24], to maximize user allocation numbers in their cost, considered the usage of edge servers to have quality of service. In addition, the data communication over the network is also considered one of the aspects of the cost.

Optimization techniques play a crucial role in the efficient management of resources in fog computing and IoT environments [30]. One popular optimization technique used in these environments is bin packing. Bin packing is a combinatorial optimization problem [31,32] that involves packing a set of items into a fixed number of bins, with the goal of minimizing the number of bins used or the overall cost of the solution. In fog computing and IoT environments, bin packing can be used to optimize the placement of services and devices, taking into account factors such as network conditions, service requirements, and device characteristics to minimize the overall cost of the solution by reducing the number of fog nodes used.

There are many variations of the bin packing problem, including the multi-dimensional bin packing problem [33] and the multi-constraint bin packing problem [34]. These variations can be used to add extra constraints and requirements in fog computing and IoT environments. For example, the multi-dimensional bin packing problem can be used to take into account the different resource requirements of services and devices, and the multi-constraint bin packing problem can be used to take into account additional constraints such as security and privacy.

In the literature, there are several works that have proposed the use of bin packing for the optimization of service allocation and task scheduling. The authors of [35] attempted to enhance task scheduling by transforming it into a bin packing problem. Three modified versions of bin packing algorithms based on the minimization of makespan were presented for use in task scheduling (MBPTS). They used the Cloudsim [36] open-source simulator. When compared to scheduling algorithms such as first come first serve (FCFS) and particle swarm optimization (PSO), the results of the proposed MBPTS were adequate to optimize balancing results, reducing the waiting time and improving resource utilization. The authors of [13] presented the edge user allocation (EUA) problem as a bin packing problem and presented a unique optimum solution based on the lexicographic goal programming technique. They ran three sets of tests to compare the suggested strategy to two sample baseline approaches. The experimental findings reveal that their strategy performs substantially better than the other two alternatives. In [37], the authors presented a methodology for minimizing resource waste through resource consolidation, which is accomplished by allocating many requests to the same machine. Bin packing is offered to perform semi-online workload consolidation. The suggested approach is built on bins, with each job allocated a bin that is subsequently allocated to a machine. The suggested approach addresses the issue of request reduction in real-time resource assignment. The suggested technique obtains information in a brief time frame, allowing for more accurate decisions. Their findings reveal that during periods of high demand, their optimal policy can result in saving up to 40% more resources than the other policies and is resistant to unpredictability in task lengths. Finally, they demonstrate that even slight increases in the permitted time window result in considerable improvements, but that bigger time windows do not always improve resource use for real-world datasets. In summary, bin packing is a powerful optimization technique that can be used to efficiently manage resources in fog computing and IoT environments. By taking into account factors such as network conditions, device capabilities, and additional constraints, bin packing can be used to minimize the number of resources used and reduce the overall cost of the solution.

The summary of existing works on resource allocation in fog computing is presented in Table 1. The authors of [38] tackle the problems of allocating resources in cloud computing by using two methods: the tasks to virtual machines distribution challenge as a linear programming model and the task allocation algorithmic solution, additionally referred to as the Hungarian algorithm-based binding policy (HABBP), to address the task distribution issue in the context of cloud computing. A genetic algorithm-based virtual machine placement (GABVMP) can also be used to address and optimize the VM deployment problem in a cloud computing setting. The authors in [39] offer the novel concept of fog-cloud clustering, aiming to address the challenge of determining the optimal number of required clusters. Their research focuses on effectively scaling the network node count within the fog environment. They present a new technique that uses mixed-integer linear programming (MILP) to derive both lower and upper constraints on the required number of clusters. The work given in [40] presents an efficient resource allocation technique known as the effective prediction-based resource allocation method (EPRAM). This technique employs both deep reinforcement learning (DRL) and probabilistic neural network (PNN) methodologies. While the DRL component handles resource allocation decisions, the PNN prediction algorithm is used to select target destinations. The proposed approach not only efficiently minimizes the time span but also improves resource consumption. In addition, another strategy used for resource allocation was adopted to enhance the RL. The study in [41] provides two significant contributions: utilizing PSO for optimization of RL hyperparameters and employing the improved RL for resource allocation in the fog environment. In terms of optimization, the hyperparameters on the improved RL present an efficient approach. In [42], the authors describe a resource allocation technique known as energy-aware multiple linear regression. The strategy focuses on balancing energy consumption and execution time, with the goal of significantly reducing both delay and response time. By considering multiple linear regressions, the proposed method successfully handles energy efficiency optimization while ensuring fast and responsive system performance. The authors in [43] conducted a thorough analysis of fog-related resource allocation using the cross-layer design concept. Their new approach takes into account the cloud center device, fog nodes, and user devices as essential aspects of the cloud–fog user ecosystem. Their main objective is to maximize efficiency and outcome by allocating resources in the most efficient way possible using a gradient-based method. The authors in [44] present an approach for displaying resources in fog computing using MEC (multi-access edge computing) APIs. These APIs make real-time information on the CPU, memory, storage, and networking capabilities of fog devices available. This useful information is used by the fog’s supervising entity to make informed decisions about distributing tasks among the network’s nodes. Furthermore, the researchers developed a Lyapunov-based optimization technique for resource allocation in fog nodes and defined the challenge of latency minimization. The authors demonstrate through simulation that their combined method, which involves both resource representation and optimization of resource allocation, efficiently minimizes latency, and hence improves system performance. Authors in [45], introduced a mathematical artifact called the Markov blanket. This artifact likely plays a significant role in the management of distributed computing continuum systems. The authors in [46] focused on addressing energy and delay constraints within a computational context. The authors proposed a strategy called transmission scheduling and computation offloading (TSCO), which aims to optimize energy consumption and reduce delays by intelligently scheduling transmission and offloading computations.

Most of the literature has focused on allocation strategies in the cloud without paying sufficient attention to service allocation in fog and IoT environments. These environments consist of various capability devices, including both constrained and high-capability devices. However, they are not as powerful as the devices in the cloud. This makes the process of allocating services to devices more challenging. It requires a sound allocation strategy while optimizing all aspects such as data communication, energy usage, resource wastage, and response time. Moreover, the proposals in the literature have focused on network conditions and device characteristics, but they have not paid attention to the technical requirements of services and tasks during the allocation process. This is important, since services and tasks have technical requirements similar to those of the devices’ capabilities. In our proposal, we considered both device capabilities and the technical requirements of services to achieve a comprehensive understanding of the allocation process and efficiently allocate services to devices.

## 3. Research Problem and Motivational Scenario

As the number of IoT devices linked to the Internet has grown, so has the number of services, and businesses have begun to install additional services for various objectives. Most IoT devices have limited resources such as RAM, CPU, and storage, along with limited battery capacity. Furthermore, each deployed service has comparable constraints related to similar resources. Additionally, it is crucial to take into account the data-processing capacities of IoT devices while implementing and distributing services. As a result, before providing information about services to IoT devices, we must understand their limitations.

Considering a hospital environment where the distribution of IoT devices and services is crucial for bothIoT and healthcare. The hospital setting is composed of many IoT devices, such as sensors, wearables, and medical devices, all of which are linked together by an IoMT system. These devices continuously monitor and collect information about patient health factors in real time. In this scenario, our focus is on service distribution within the IoMT system to ensure efficient resource utilization and provide excellent patient care. Each patient who visits the hospital is given a unique identification, and their health information is collected and securely preserved. In addition, the physician allocates each individual a priority rating according to the seriousness and necessity of their health condition.

Whenever a patient comes to the hospital, they are subjected to a range of tests and examinations to determine the severity of their illness. These examinations can range from simple blood tests and an electrocardiogram (ECG) to more complicated diagnostic scans like magnetic resonance imaging (MRI) or X-rays. Each test generates various types of data with varying levels of complexity and resource requirements. For example, MRI and X-rays produce videos and images data in healthcare. Additionally, the physician determines whether the patient is of high or low priority. The system then treats this process as a single service, considering its priority. These services with priority levels will be allocated to the fog devices ($FD_{1}$, …, $FD_{n}$) for processing. The fog devices will start by processing the high-priority services, followed by the processing of lower-priority services. Finally, when the fog devices are unable to process the services (either due to low or high priority), the cloud devices ($CD_{1}$, …, $CD_{n}$) will receive the services for further analysis. For example, if a patient is listed as a high priority, then the system will send the patient’s service to the fog device for processing and obtaining a fast response. However, if the patient is in low-priority condition and the fog devices are not capable, then the service will be allocated to the cloud. In only one situation, when the fog devices are not capable of processing high-priority services, they will be allocated to the cloud for further processing.

Figure 1 depicts various types of fog, IoT, and healthcare equipment within a hospital environment. The hospital contains numerous patients ($P_{1}$, …, $P_{n}$) who are undergoing health check-ups. Moreover, the hospital offers a variety of services ($S_{1}$, …, $S_{n}$) to patients. Some patients are seeking treatment for dental problems, immunization, lung issues, kidney and internal medicine problems, diabetes, eye problems, brain disorders, pregnancy, and heart conditions. In our scenario, we regard these health issues that patients experience as components of services (S). In other words, each user can have one or more services. When a user has a service, the service ($S_{i}$) will contain all the patient and test data.

Furthermore, analyzing image and video data may require more processing capacity than numerical analyses like blood test results or ECG results. This is due to their volume and the methods they employ, indicating that the capabilities of the fog layer must be sufficiently robust to handle these services. As a result, because fog devices have limited processing capability when compared to cloud devices, it is not viable to implement these services on them. Due to resource-constrained devices in regard to hardware, service allocation is a crucial part of fog architecture. Furthermore, certain fog devices remain unused due to their limited power capabilities for service execution. This implies that although these fog devices are required for data processing, they are overlooked due to power limitations. This situation can worsen when billions of services are sent to numerous devices and executed by them. This signifies that there is a waste of network devices, which may cause the computation time to be delayed. The waste of devices occurs when devices are not used owing to restrictions and remain in the network unused, losing the available resources. The main challenge is determining which service should be allocated to which device in a fog architecture while maintaining overall efficiency. This is analogous to optimization problems, which can be deemed highly suitable for allocation problems.

### 3.1. Assumptions

We assume that the fog layer’s devices have limited RAM capabilities compared to cloud devices. For our experiments, a synthetic dataset is generated. The dataset has services’ technical requirements, fog devices’ capabilities, and the priorities of services. The dataset has 800 services with varying technical requirements, and there are 50 fog devices with varying capabilities. The technical needs of services and the devices’ capabilities are known. The connection between devices in our experiments is outside the scope of our study.

### 3.2. Process

The model starts by building synthetic data for both the requirements of services and the capabilities of devices to prepare them for the allocation model. The fog devices’ capabilities are predefined, with fog devices being less capable when compared to cloud devices. The allocation technique is used to distribute services based on service requirements, service priority, and device capabilities. Services will be allocated across fog or cloud devices depending on the needs of the services, considering device capabilities and service priority.

## 4. Methodology

We propose the PSA and SSA techniques, which are techniques for allocating resources based on a list that includes all fog devices connected to the network. This method allows us to determine the sequence in which different services are executed by various devices. The actual capacities of devices, such as RAM, are used to arrange the list of devices. As a result, for each physical aspect, all devices’ capabilities are maintained in a list. The main purpose is to allocate services to devices in an effective and optimized manner. The allocation technique is utilized to allocate all or a part of the services to a specific number of fog devices or to various cloud devices with varied capabilities according to their capability. Furthermore, the allocation approach aims to maximize fog device usage and minimize data communication over the network. An overview of our proposed strategy is presented below with equations.

The main goal *G* is to allocate the services $s_{i}$ to fog devices $D_{F}$ as much as possible in an effective manner. We can represent this as:$$\max_{D_{F}}\sum_{i=1}^{N}s_{i}\cdot A_{i,F}$$
where $A_{i,F}$ is a binary variable that represents whether or not service $s_{i}$ is allocated to fog device $D_{F}$. If $A_{i,F}=1$, then service $s_{i}$ is allocated to fog device $D_{F}$, and if $A_{i,F}=0$, then service $s_{i}$ is not allocated to fog device $D_{F}$. The notation $\sum_{i=1}^{N}s_{i}\cdot A_{i,F}$ calculates the total number of services allocated to fog device $D_{F}$, where *N* is the total number of services. The objective is to maximize this quantity over all possible allocations to fog devices.

The services will be allocated by the fog device to either fog devices $D_{F}$ or cloud devices $D_{C}$, dependent on the capabilities of the devices and the computing needs of the services. We can represent this as:$$s_{i}\to \left\{\begin{array}{cc} D_{F}\left(s_{i}\right), & \mathrm{if}\;D_{F}\;\mathrm{can}\;\mathrm{handle}\;s_{i}\\ D_{C}\left(s_{i}\right), & \mathrm{otherwise} \end{array}\right.$$

The services will then be allocated according to their requirements to the fog devices since this is the focus of our strategy. We can represent this as:$$s_{i}\to D_{F}\left(s_{i}\right)$$

If the fog devices are unable to manage the load, the remaining services will be allocated to the cloud-based devices. We can represent this as:$$s_{i}\to D_{C}\left(s_{i}\right)$$

Therefore, to have efficient results, we used both fog devices and cloud devices for service allocation. We can represent this by combining the previous two relations as follows:$$D_{F}\left(s_{i}\right),D_{C}\left(s_{i}\right)\to G\left(s_{i}\right)$$

### 4.1. Objective Function

We have a multi-criteria optimization problem. The goal is to maximize the weighted sum of two objective functions, $f_{1}\left(x\right)$ for decreasing delay and $f_{2}\left(x\right)$ for optimizing the usage of resources. The weighting variables $w_{1}$ and $w_{2}$ are used to balance the significance of the two objectives and manage the trade-off between conflicting goals. These can be considered performance objectives for our strategy. The function that needs to be maximized is described below:(1)$$\mathrm{Maximize}\;f_{m}\left(x\right)=-w_{1}\cdot f_{1}\left(x\right)+w_{2}\cdot f_{2}\left(x\right)$$
where $f_{1}\left(x\right)$ is the objective function for minimizing latency (i.e., the time it takes for a service to be allocated from fog to cloud), and $f_{2}\left(x\right)$ is the objective function for maximizing fog resource utilization (i.e., using the fog devices as much as possible while minimizing resource wastage). Here, *x* is a vector of variables, and $w_{1}$ and $w_{2}$ are weighting factors used to balance the importance of the two objectives.

The minus sign in front of $w_{1}\cdot f_{1}\left(x\right)$ indicates that we are maximizing the negative of $f_{1}\left(x\right)$, which is equivalent to minimizing $f_{1}\left(x\right)$. Similarly, we maximize $f_{2}\left(x\right)$ by multiplying it with a positive weight $w_{2}$. The aim of integrating two objectives into a single objective function is to determine the optimal trade-off between them. By maximizing $f_{m}\left(x\right)$, we intend to find *x* values that simultaneously minimize $f_{1}\left(x\right)$ and maximize $f_{2}\left(x\right)$, with appropriate weightings. The range of values of $w_{1}$ + $w_{2}$ = 1.

#### 4.1.1. Best Fit

The presented algorithm, called “Best Fit”, is used to allocate services to devices depending on their capacity and technical requirements of services. We provide the best fit code in Algorithm 1 to maximize device usage while delivering services to devices based on their capabilities. We require the service requirements serReq and device capabilities devCap as input. Then, for each service, we find the smallest possible device capability that may accommodate the current service.
(2)$$Service_{present}=\mathrm{find\_min}\left(devCap_{1},devCap_{2},\ldots ,devCap_{n}\right)$$

If a device is found, it should be assigned to the current service. If a device cannot be found, we disregard it and proceed with the other services. We do not break down the services into smaller ones, but rather assign them to one of the devices, either a fog device or a cloud device, according to their capabilities.
**Algorithm 1** Best Fit**Input:** devCap[],
serReq[]**Output:** allocID[]  **for** x←0 
**to** 
length(serReq)−1 
**do**     bestFitID←−1     **for** y←0 **to** length(devCap)−1 **do**       **if** devCap[y]≥serReq[x] **then**         **if** bestFitID=−1 **then**             bestFitID←y         **else if** devCap[bestFitID]>devCap[y] **then**             bestFitID←y         **end if**     **end if**   **end for**   **if** bestFitID≠−1 **then**     allocID[x]←bestFitID     devCap[bestFitID]←devCap[bestFitID]−serReq[x]   **end if****end for**


In other words, services are assigned to available devices based on the best fit criteria. The algorithm takes two parameters: devCap and serReq. Each device’s capacity is represented by devCap, and the service request is represented by serReq. The algorithm begins with two nested for-loops. The outer loop starts at 0 and iterates up to the length of serReq, while the inner loop starts at 0 and goes up to the length of devCap. The outer loop processes every service request one at a time, while the inner loop checks every device’s capacity to see if it can handle the present service request.

The bestFitID is initially set to −1. If the capacity of the device at index *y* is greater than or equal to the capacity of the service request at index *x*, the bestFitID is assigned to *y* in the inner loop. If bestFitID remains −1, it signifies that no device is currently allocated to the service. If bestFitID has been assigned a value, a comparison is made between the present device’s capacity (devCap[y]) and the device assigned previously (devCap[bestFitID]). If the current device’s capacity is smaller than that of the earlier assigned device, the bestFitID is changed to the present device (*y*).

After the inner loop completes, if bestFitID is not −1, the current service is assigned to the device with the best fit (bestFitID). The allocation is recorded by updating the allocID list, and the capacity of the device is decreased by the service request. The algorithm will keep running the outer loop until all the service requests are processed, and the devices are allocated to the services. The method returns allocID, which represents the index of the device allocated to the service. The result of the algorithm is the allocation of services to the devices.

For instance, let us consider the following fog device capabilities: FDC1 = 16,384 MB, FDC2 = 8192 MB, FDC3 = 4096 MB, and FDC4 = 2048 MB; along with the following service requirements: 2048 MB, 2048 MB, 2048 MB, 2048 MB. Table 2 shows the results of best fit example after running the code.

Service request = 2048, allocation ID = 1, fog device = 2048. This indicates that the first service requirement, which has a size of 2048 units, is assigned to a fog device with a RAM of 2048.

Service request = 2048, allocation ID = 2, fog device = 4096. The second service request, also of size 2048 units, is assigned to a different fog device with a RAM of 4096. Because this device has a capacity of at least 2048, it can accommodate the service request.

Service request = 2048, allocation ID = 3, fog device = 4096. The third service request, also with a size of 2048 units, is assigned to the same fog device as the second request. The fog device’s capacity is still adequate to handle this request.

Service request = 2048, allocation ID = 4, fog device = 8192. The fourth service request, also with a size of 2048 units, is assigned to a different fog device with an ID of 8192. This device has enough capacity to accommodate the request.

In summary, the allocation process involves assigning each service request to a specific fog device based on available capacity. The allocation IDs help track which request is assigned to which fog device. The process ensures that fog devices with sufficient capacity are selected to handle the requests. As the allocation progresses, the capacity of the fog devices is adjusted accordingly.

#### 4.1.2. Worst Fit

We provide the worst fit code in Algorithm 2 to optimize device usage while assigning services to devices and considering device capabilities. We require service requirements serReq and device capabilities devCap as input. Following that, we select each service and identify the most capable device that can support the current service.
(3)$$Service_{present}=\mathrm{find\_max}\left(devCap_{1},devCap_{2},\ldots ,devCap_{n}\right)$$

If a device is found, it should be assigned to the current service. If a device cannot be found, ignore it and continue investigating the other services.

The method then iterates through each service request in the array serReq. It changes the value of worstFitID to −1 for each service request, indicating that no device has been allocated to the service yet. Next, the method iterates over the device capacities in the array devCap. If a device has adequate capacity to satisfy the current service request, the algorithm determines whether it is the first device discovered with sufficient capacity or if its capacity is more than the current worstFitID. If the device’s capacity is greater, the algorithm assigns worstFitID to the present device’s ID.

Following the completion of the inner loop, the algorithm checks if a device has been allocated to the current service requirement. If a device is allocated, the algorithm updates the allocID list by distributing the current service requirement the value of worstFitID. The algorithm also decreases the allocated device’s capacity by the magnitude of the service request. The aforementioned stages are repeated by the algorithm for all service requests. After the outer loop has been completed, the algorithm returns the allocID list, which contains the list of allocated services to devices.

The worst fit algorithm allocates the services to devices to ensure that each service is allocated to the fog device with the most available capacity, maximizing resource consumption in a multi-device scenario.
**Algorithm 2** Worst Fit**Input:** devCap[],
serReq[]**Output:**  allocID[]  **for** 
x←0 
**to** 
length(serReq)−1 
**do**     worstFitID←−1     **for** y←0 **to** length(devCap)−1 **do**       **if** devCap[y]≥serReq[x] **then**          **if** worstFitID=−1 **then**                worstFitID←y          **else if** devCap[worstFitID]<devCap[y] **then**                worstFitID←y          **end if**       **end if**     **end for**     **if** worstFitID≠−1 **then**          allocID[x]←worstFitID          devCap[worstFitID]←devCap[worstFitID]−serReq[x]     **end if**  **end for**


#### 4.1.3. First Fit

To maximize device utilization while allocating services to devices based on their capabilities, we present the first fit code in Algorithm 3. We need service requirements serReq and device capabilities devCap as input. After that, we select a service and determine whether it is compatible with the current device. If the devCap is equal to the serReq, we assign it and proceed to the next serReq. If not, we move on to investigate the next devCap.
**Algorithm 3** First Fit**Input:** devCap,
serReq**Output:**  allocID[]  **for** 
x←0 
**to** 
length(serReq)−1 
**do**     **for** y←0 **to** length(devCap)−1 **do**       **if** devCap[y]≥serReq[x] **then**          allocID[x]←y          devCap[y]←devCap[y]−serReq[x]          **break**       **end if**     **end for**  **end for**


In other words, the method has two inputs: devCap (device capabilities), serReq (service needs); and one output: allocID (allocated IDs). The algorithm’s purpose is to allocate services in serReq to devices in devCap and record the allocation in allocID.

The algorithm begins with two nested for-loops, where *x* is the length of serReq and *y* is the length of devCap. The method examines whether devCap[j] is larger than or equal to serReq[x] in every iteration of the inner loop. If the condition is met, the algorithm assigns the present service (serReq[x]) to the current device (devCap[j]) by changing allocID[x] to *y* and lowering the current device’s capacity by the service requirement (devCap[j]−=serReq[x]). Finally, the algorithm exits the inner loop after locating a device capable of allocating the current service and proceeds to the following service in the outer loop. This operation is repeated by the algorithm until all services are assigned to devices.

#### 4.1.4. Priority-Based Service Allocation

We present the code for the priority-based allocation in Algorithm 4 to select the service allocation process and if the services should be handled in the fog or cloud. We need service requirements serReq and device capabilities devCap as input. The services are then allocated to the fog devices using the fogDevice(devCap,serReq) method. Following that, one of the algorithms presented previously will be executed. Following that, we will evaluate the capabilities of fog computing in order to allocate services. If no fog devices are available to handle the service, we assign it to cloud devices by calling and forwarding the remainder of the services to cloudDevice(remainingSer).
**Algorithm 4** Priority-Based Allocation**Require:** devCap: list of available devices, serReq: list of service requirements, allocID: list of allocation IDs**Ensure:** allocID1: remainingSer← empty list2:  **for** each service request *x* in serReq **do**3:   **if** priority of serReq[x] is high **then**4:     Allocate to fog devices with sufficient capacity5:     **if** allocID[x]≠−1 **then**6:        Increment allocID[x] by 17:     **else**8:        Append serReq[x] to remainingSer9:       **end if**10:   **else if** priority of serReq[x] is low **then**11:     Allocate to fog devices with sufficient capacity12:     **if** allocID[x]≠−1 **then**13:        Increment allocID[x] by 114:     **else**15:        **if** there is sufficient capacity in fog devices **then**16:             Allocate serReq[x] to a fog device17:        **else**18:             Append serReq[x] to remainingSer19:        **end if**20:     **end if**21:   **end if**22:** end for**23: **if** remainingSer is full **then**24:   Allocate remaining services to cloud devices25: **end if**


Based on the importance of the service request, the algorithm decides whether to distribute services to fog or cloud devices. The algorithm prioritizes service requests by initially assigning them to the fog devices. The algorithm begins the process by determining the level of priority of the service demand. If the service is high-priority, the algorithm inputs the device capability and service requirement. If fog devices are capable of handling the services, they will be allocated to fog devices. If fog devices are unable to handle the low-priority service, the algorithm will allocate it to cloud devices instead.

The algorithm begins by allocating services to fog devices. To distribute services to fog devices, the algorithm employs one of three allocation algorithms: best fit Algorithm 1, worst fit method Algorithm 2, or first fit Algorithm 3. The method then iterates across the length of the service request using a for loop. In every iteration, the algorithm determines whether or not the service has already been assigned to a device by determining whether or not the allocID is greater than −1. If the allocID is not equal to −1, it is increased by one. By incrementing the allocID in such cases, the algorithm ensures that each service request allocated to fog devices receives a unique identifier, preventing conflicts and ambiguity in the allocation process. If the allocID is −1, the service request is saved in the remainingSer list.

The algorithm checks if the remainingSer list is empty at the end of the for-loop. If the remainingSer list is empty, the algorithm has been completed and all services have been allocated to the devices. If the remainingSer list is not empty, the method “cloudDevice” with the remainingSer list as input is called. The “cloudDevice” function is in charge of distributing the remainder of services to cloud devices. The priority-based allocation mechanism routes service requests to either the fog or cloud layers based on their priority. The method assigns services to fog devices utilizing one of three allocation techniques: best fit, worst fit, or first fit, with the remaining services given to the cloud.

#### 4.1.5. Sort-Based Service Allocation

We employed dual-pivot quicksort, an efficient sorting algorithm commonly used in computer science and data-processing applications, especially for sorting primitive data types such as int, double, and float. Dual-pivot quicksort, according to the authors’ findings [47], is typically quicker and more effective than alternative quicksort algorithms, especially on big datasets. They point out that dual-pivot quicksort works effectively with both randomly ordered and partly sorted data, and that it has a minimal amount of comparisons and swaps.

Sorting may be a valuable technique in the context of fog computing for optimizing service allocation and lowering the level of complexity of service distribution for fog devices. Sorting the data prior to them reaching the fog devices makes it easier to deploy resources and maximize the network’s overall performance. We classify the technical needs of services in ascending order, from lowest to greatest, to assist service allocation techniques for fog devices. We also sort the capabilities of fog devices. This allows for faster and more efficient service allocation.

### 4.2. Architecture

Our architecture is organized into three major sections: a sensor layer, a fog Layer, and a cloud layer. First, the sensor layer has IoMT sensors and IoMT devices that send data to the fog layer. Second, the fog layer is responsible for distributing services to devices in an effective and optimized manner by ensuring that all available resources are utilized and serving users by offering accessible services. Last, the cloud layer can manage all of the data and services, as well as provide essential services to the edge and fog layers.

The architecture shown in Figure 2 incorporates three devices, including IoMT, fog, and cloud, as shown below:The IoMT sensors and devices are situated in the sensor layer of the network system and are typically integrated into real-world objects. IoMT sensors are small and cost-effective, making the installation process straightforward and cost-efficient. These devices communicate with the fog devices using wireless communication. The IoT ecosystem should be beneficial in a variety of ways, including energy savings, lower costs, better resource use, and lower data transmission costs via the network. The IoMT sensors and devices generate data for each patient, which are then integrated into the service. This ensures that each service possesses data about the patient and their medical diagnosis. Then, after fusing all the data into services, the services will be sent to the fog layer. Additionally, this layer is responsible for sorting and prioritizing services, aiding the fog layer in allocating services to devices.The fog devices reside adjacent to the sensor layer or within the communication channel and gather services and use a service allocation strategy to allocate the services to the fog devices as the priority is to process the services closer to the data source. However, whenever the fog devices cannot handle the services due to the lack of power of fog devices, then the services will be sent to the cloud using the proposed allocation strategy. Additionally, the fog devices are responsible for allocating services to either fog or cloud devices based on the priority of the service. Clearly, fog devices have very little power and a narrower global data perspective than cloud devices; thus, they can store less data and provide fewer services.The cloud devices receive services from fog devices. The computational power and data storage capacity of the cloud are significantly greater than those of fog devices and IoMT devices. Cloud devices can be used for further analysis and storage when required to have the full picture of the data.

## 5. Experimental Setup

### 5.1. Dataset

In this study, we performed experiments to evaluate the effectiveness of fog computing for service allocation. We utilized a customized dataset comprised of several fog devices configurations and services settings to conduct our research. The dataset used in this study was created in order to simulate an IoMT healthcare system.

#### 5.1.1. Fog Devices’ Configurations

The fog device configurations utilized in the experiments are detailed in Table 3. The table displays the experiment name, the number of fog devices, and the fog devices’ RAM capacities (in Gigabytes). The experiment name is divided into three sections: (1) the allocation approach (worst fit, best fit, or first fit), (2) the configuration types (low, medium, or high), and (3) the device capabilities (FDC1 to FDC4). Each method includes three tests of varying configurations (low, medium, and high).

FDC1, FDC2, FDC3, and FDC4 are the fog devices utilized in the experiment, each with 2 GB, 4 GB, 8 GB, and 16 GB RAM. The low configuration had 50 fog devices totaling 100 GB; the medium configuration had 15 FDC1, 15 FDC2, 10 FDC3, and 10 FDC4 fog devices totaling 330 GB; and the high configuration had 50 FDC4 fog devices totaling 800 GB.

#### 5.1.2. Service Setups

The technical requirements of the services utilized in the experiment are provided in Table 4. The services are labeled SR1, SR2, SR3, and SR4, with RAM requirements ranging from 1 MB to 2 GB. We assumed that the size of the services matched the technical requirements of the services in terms of GB size. We deployed these services in various configurations to assess the performance of fog computing under various circumstances. A total of 800 services with varied technical needs were generated, and the generated data were used in all experiments. Half of the services (400) have high priority and the other half (400) have low priority. To create the dataset, simulations were run with fog devices configured as stated in Table 3 and services with varying technical requirements shown in Table 4. The data were gathered and evaluated in the study. As the dataset was generated at random, it was representative of real-world scenarios and provided various types of data for analysis. The dataset was used to assess the influence of various allocation techniques on system performance and resource consumption in an IoT healthcare system.

This arrangement was created to emulate real-world circumstances in which fog devices of diverse capacities may be required to host a variety of services with varying resource requirements. The research intended to evaluate and compare several methods for service allocation in fog computing environments by varying the number and capacity of fog devices as well as the resource requirements of the services. Overall, the dataset utilized in the experiments offers a wide range of fog device and service configurations for evaluating fog computing performance.

### 5.2. Experiments

We categorized the experiments into three categories: those without priority and sort (standard), those with priority, and those with sort. Experiments with no priority or sort are used to allocate services without consideration for characteristics like priority or sort. The experiments with priority concentrate on the priority considerations when distributing services to fog devices; this indicates that services with a high priority will be delivered to fog devices first, followed by those with low priority. When the fog is lacking, the services will be assigned to cloud devices. The experiments with sort initially sort the services in the sensor layer from small to big depending on their requirements in order to make the allocation process for fog computing easier and to support the algorithms in distributing the services efficiently. The three categories of experiments are employed to evaluate our allocation strategy. When a setup does not necessitate specific factors, the first strategy is chosen. If a setup requires prioritization, experiments with a priority focus are selected, as we aim to deploy services to fog devices whenever possible.

We performed a total of three main experiments in each category of experiments. In every category (standard, priority, and sort), we used three algorithms with three different configurations, namely low, medium, and high, as mentioned earlier. In total, we conducted nine experiments for each category to discover the ideal configuration for fog devices in order to distribute services as effectively as feasible across fog devices. The configuration of fog devices may differ based on the experiment, as illustrated in Table 3. The first column lists the titles of the experiments, while the second column lists the capabilities of the fog devices. We deployed 800 services using a variety of capabilities (FDC1, FDC2, FDC3, and FDC4) to each of the 50 fog devices in experiments 1 through 9. The broad range of capabilities includes both highly specialized and relatively common equipment (running all capacities).

## 6. Results

In this section, we explore the outcomes of several techniques for allocating services to fog or cloud depending on priority, size, and algorithms employed. The outcomes of the allocation approach are shown here. As previously stated, the technique would first allocate services to fog nodes based on their capabilities and then assign services that could not be managed in the fog to cloud devices. We will examine and provide the findings of the three methods, namely worst fit, best fit, and first fit. According to Statista [48], the average upload speed for transmitting services from fog to the cloud using mobile is 8.5 Mbps. However, the average upload speed using fixed broadband to transfer services from fog to the cloud is 28.5 Mbps.

To begin, we give two charts in Figure 3 and Figure 4 that illustrates the distribution of high and low-priority services to fog or cloud using various methodologies. Standard, priority, and sort are the strategies used in our experiments. The allocation is provided separately for high- and low-priority services. Second, in Figure 5a,b, we provide a bar chart that indicates the number of unused fog devices in GB after assigning services to fog. Furthermore, the values in the table represent the amount of unused RAM in GB for each algorithm and each level of fog device configuration: low and medium. In the charts, we did not show the results of high level configuration of fog devices as all the services were handled in the fog. Then, in Figure 6a,b, we show a chart and a table that provide information about allocating services to fog or cloud using three different strategies: standard, priority, and sort. The services are categorized as high- or low-priority, and their sizes are indicated in gigabytes (GB). Finally, we talk about the time it takes for services to travel from fog devices to the cloud, considering a variety of techniques.

The findings are reported in terms of the time required to assign services via mobile and broadband networks, and the time is determined based on the upload speed supplied by Statista [48].

Figure 3 and Figure 4 display the quantity of high- and low-priority services assigned to fog or cloud using various techniques, including standard, priority, and sort. The allocation is displayed separately for high-priority and low-priority services. “Standard” refers to a situation in which there is no differentiation between high- and low-priority services. Priority denotes a situation in which high-priority services take precedence over low-priority services. Sort denotes a scenario in which services are first sorted and then assigned using the best fit, worst fit, or first fit algorithm. According to the findings, the allocation approach has a substantial influence on the number of services provided to fog and cloud environments. In general, the priority method results in more high-priority services being assigned to the fog environment and more low-priority services being assigned to the cloud environment. When it comes to sorting services, the allocation approach has less of an influence on the number of services distributed to the fog and cloud environments. However, the allocation algorithm utilized does make a difference. For example, the worst fit algorithm allocates more services to the fog environment, but the first fit algorithm allocates more services to the cloud environment.

For each combination of distribution strategy and service priority, the chart illustrates the number of distributed high and low-priority services. The charts show that the number of high-priority services allocated to the fog is lower than that in the cloud for the standard strategy, while the converse is true for low-priority services. We did not include the results of high-capacity fog devices in the table since their exceptional capabilities allowed them to manage all services in the fog as mentioned earlier. It is clear from the charts that all algorithms are doing well in terms of allocating services to fog and cloud, but as our intention was to push the processing near the data source, a priority strategy can be a good choice. Among the three algorithms, we can realize that the best fit can be selected as the best in most cases.

Overall, our findings imply that a careful evaluation of allocation methods and algorithms could be useful in optimizing service allocation in fog and cloud situations. A priority approach, in particular, that prioritizes high-priority services, can help guarantee that important services are allocated to the fog environment, where they can be handled quickly and efficiently. Whereas the standard and sort strategies appear inefficient since they allocate services without regard for priority.

The number of distributed services (NDS) is calculated using the following equation: it is a metric for determining the proportion of services delivered or allocated to fog devices within a given configuration.
$$\mathrm{NDS}=\left(\frac{\;\mathrm{Number}\;\mathrm{of}\;\mathrm{services}\;\mathrm{using}\;\mathrm{strategy}}{\;\mathrm{Total}\;\mathrm{number}\;\mathrm{of}\;\mathrm{services}\;\mathrm{in}\;\mathrm{configuration}\;}\right)\times 100$$

The resultant number represents the proportion of services that are effectively distributed or allocated to fog devices within the given configuration based on the specified strategy. A higher score indicates that more services have been successfully allocated. The results are presented in Table 5 and Table 6.

The bar charts in Figure 5a,b illustrate the number of unused fog devices in GB after assigning services to fog. The values in the table indicate the quantity of unused RAM in GB for each algorithm and fog device configuration: low, medium, and high. Worst fit low: This allocation approach directs resources to the fog gadget that produces the most waste. According to the chart, this technique wastes 2 GB for devices with low capacity within the standard strategy. When priority is set, the wasted space is reduced to 1.5 GB. Worst fit medium: This allocation approach produces the largest waste among medium- and low-configuration devices. According to the charts, this technique wastes 7 GB for devices in standard and 6 GB with priority.

However, when medium-configuration devices are sorted, the strategy wastes 42 GB. Best fit: This allocation strategy allocates resources to the fog device with the least wastage. The table shows that this strategy results in 0 GB of wastage for devices in all strategies. First fit low: This allocation strategy allocates resources to the first available fog device. The table shows that this strategy results in 1.5 GB of wastage for devices with low capacity within the standard and 6 GB within the sort strategy, but 1 GB within the priority strategy. First fit medium: This allocation strategy allocates resources to the first available fog device among devices with medium capacity. The table shows that this strategy results in 2 GB of wastage for devices within the standard and 1.5 GB of wastage when priority is given; however, the wastage increases to 6 GB when the sort strategy is used.

Overall, we did not reveal the high-config devices since there was no waste because the fog devices had high capabilities, which allocated all services to the fog devices. Finally, the table shows that the Worst Fit strategy results in the most waste for both low- and medium-capacity devices, especially when the sort is provided. In all circumstances, the best fit technique results in the least amount of waste. The first fit approach stands somewhere between the other two.

The charts in Figure 6a,b show the RAM size of services allocated to fog or cloud based on three alternative strategies: standard, priority, and sort. The services are classified as high- or low-priority, and their volumes are measured in gigabytes (GB). The low-configuration devices have a total of 100 GB and the medium-configuration devices have a total of 330 GB, as mentioned earlier. This indicates that in each configuration of low, medium, and high experiments, there is a maximum size of RAM to handle services. For example, the worst fit low experiment allocated 98 GB of services to fog devices of that size out of 100 GB; the remaining 2 GB is the wastage discussed earlier. It is clear that the strategy used most of the fog devices’ RAM, but most of the services traveled to the cloud because of the limited capabilities of the fog devices. It is clear that the best fit low and medium strategies are the best among others in terms of the usage of fog devices and allocating all services that can fit the fog devices without wastage. In the low configuration, the sort strategy stands better than the standard strategy in worst fit and best fit, but worse in sort. In the medium configuration, the sort strategy has better results than the standard in all algorithms. Overall, the table provides valuable information about the allocation of services to the fog or cloud and highlights the importance of considering service size, algorithm, and strategy when making these allocation decisions.

Table 7 illustrates how long it takes for services to travel from fog devices to the cloud using various techniques. The algorithms employed were standard, priority, and sort. The findings are reported in terms of the amount of time required to distribute services using mobile and broadband networks. The data clearly show that the time required to distribute services to the cloud is often longer for the mobile network than for the broadband network. This is most likely due to the fact that the mobile network has more constraints and requirements for service distribution than the broadband network. In terms of the various algorithms utilized, we can observe that the best fit algorithm outperformed the others for both mobile and broadband networks. For the mobile network, the worst fit algorithm performed the worst, but it was comparable with the other methods for the broadband network. The first fit method performed effectively in the broadband network but not so well on the mobile network. It is clear that the high-configuration setup of fog devices had no cost over the network as all the services were handled locally. Overall, the table findings indicate that the best fit algorithm may be the most effective method for distributing services from the fog to the cloud. However, the particular method used may be determined by the network’s specific requirements and constraints.

The following equation illustrates the data communication reduction (DCR) in hours. Time_of_services_using_strategy is obtained from the Figure 6a,b. The results of enhancement are presented in Table 8. In addition, time of total services without strategy is obtained from Table 9 and the data are used in the equation.
$$\mathrm{DCR}=100-\left(\frac{\mathrm{Time\_of\_services\_using\_strategy}}{\mathrm{Time\_of\_total\_services\_without\_strategy}}\right)\times 100$$

For example, the time it takes for services to travel between the fog and the cloud in worst fit low using mobile in standard strategy is 3 days, 7 h, and 45 min. This should be converted to hours, and when it is converted to hours, it will be 79.75 h out of 107 h in total. Then, this number will be used in the equation.

### 6.1. Total Services Allocated to the Cloud

In this experiment, we allocated all services to the cloud in order to compare our techniques against allocation without a strategy and to investigate architecture-based allocation. Table 9 clearly shows that “No of Services” indicates 0 services out of a total of 800 services assigned to fog devices and 800 services out of 800 services assigned to cloud computing. In other words, the fog devices receive 0% of the allocated services, while the cloud devices receive 100% of the services. The column “Allocated Services in GB” in the table indicates that 0 services of 0 GB in size are allocated to the fog and 383 GB are allocated to the cloud devices. The time of service allocation is then represented as “Time of service allocation (mobile)”, which indicates that the amount of time it takes for a mobile network to allocate the services to fog is 0 due to no costs associated with local data communication. but it takes 4 days, 11 h, and 0 min to allocate the 383 GB of services from fog to cloud using the mobile network. However, “Time of service allocation (broadband)” means that the period of time it takes to allocate services from fog to the cloud using the fixed broadband network is 1 day, 1 h, and 40 min.

Table 10 shows the execution time of each algorithm in three strategies, namely standard, priority, and sort. The most effective strategy is the sort strategy; when compared to the standard and priority methods, the sort strategy consistently has shorter execution times. The sort strategy involves a sorting technique, which optimizes resource allocation and results in faster algorithm execution times. As a result, the sort strategy is the optimal method for reducing execution times and optimizing resource allocation efficiency. However, without any priority or sorting factors, the standard method often has longer execution durations than the other techniques. In terms of optimizing resource allocation efficiency, it lacks the benefits of priority levels and sorting approaches. As a result, the standard approach is the least effective in terms of execution times and resource allocation efficiency.

The priority strategy may outperform the standard strategy in terms of resource allocation efficiency. In terms of reducing execution times and optimizing resource allocation efficiency, the Sort method outperforms both the priority and sort strategies. In summary, the sort strategy is the optimal strategy in terms of execution time, since it reduces execution times. The standard method is considered the least favorable since it lacks the advantages of priority and sort, while the priority approach can occasionally give better resource allocation efficiency than the standard strategy.

We conducted three experiments using three methods, namely best fit, first fit, and worst fit. Based on the results, the worst fit results were lower than the best fit and first fit; it is clear that the data communication over the network is reduced with all strategies when compared to total service allocation to the cloud, and most of the services are allocated to the fog devices. Since reducing data transfer across the network was our primary goal, it fits with our proposed technique. Furthermore, the second aim was to use the priority aspect to allocate the services as much as possible to the fog devices, and based on the results, this aim was achieved, as 90% or more of the services were allocated to the fog devices.

We conducted a total of twenty-seven experiments using the best fit, first fit, and worst fit methodologies. According to the results, the worst fit and first fit results were lower than the best fit, indicating that data traffic via the network decreased with all methods when compared to the entire service allocation to the cloud, with the fog devices receiving the majority of the services. It fits with our proposed technique because reducing data transfer across the network was our primary goal. Furthermore, the second goal was to use the priority aspect to allocate as many high-priority services as possible to fog devices, and based on the results, this goal was achieved because 90% or more of the priority services were allocated.

We compared the performance of the algorithms, namely best fit, worst fit, and first fit, while allocating services to fog devices based on the variable capability of fog devices and variable service requirements. The focus was on the number of services allocated to fog, resource usage, and data communication over the network. The first fit algorithm allocates services to the first fog device that can handle them. It may, however, produce some wastage, lowering fog device usage. The use of fog devices was approximately 90%. The best fit algorithm allocates services to the smallest fog device that can handle them. This algorithm tries to reduce waste and maximize fog device use. The wastage was found to be the lowest. The use of fog devices in the priority strategy was 100%, and in other strategies, it was more than 90%. The worst fit algorithm allocates services to the largest available fog device that can handle them. However, this algorithm leads to increased wastage. The wastage was found to be the highest. Additionally, the fog device usage was better than the first fit, and better than the best fit in some of the experiments, but generally worse than the best fit. In terms of waste, the best fit algorithm outperformed the first fit and worst fit, attaining the lowest wastage. The worst fit algorithm generated the most wastage. As a result, the best fit algorithm is regarded as the most advantageous of the three for the variable capability of fog devices and variable services requirements scheme, since it reduces waste and maximizes fog device use. Based on our knowledge, the worst fit can lead to high wastage (fragmentation) and the best fit can be best in terms of wastage. However, in some cases, the worst fit can increase the usage of resources [49].

## 7. Discussion and Evaluation

We used two commonly used evaluation measures to assess the effectiveness of our model: the allocation success rate (ASR) and the average resource usage (ARU). The ASR calculates the number of services successfully assigned to fog devices/cloud devices. The ARU calculates the percentage of RAM used by all fog devices/cloud servers. To illustrate the robustness of our model, we conducted sensitivity analysis by adjusting various factors such as dataset size, technological requirement distribution, and the number of fog devices/cloud servers. The findings indicate that our model is not overfitting a specific dataset or set of parameters. We compared our strategies, including standard, priority, and sort performance, while considering commonly used algorithms in the literature: the best fit, first fit, and worst fit algorithms. Our model performs well in terms of ASR and ARU, according to the results. We used various data groups randomly generated with different distributions and sizes to demonstrate the validity of our model. The findings reveal that our proposal performs well with the given setups and distributions. Our methodology proposes a realistic and efficient solution to the service allocation problem in fog computing applications.

Based on the results, we made the following observations:

### 7.1. The Number of Services Allocated to Fog Devices

As previously said, our primary aim in fog computing is to allocate services as close to the data as possible while simultaneously maximizing resource consumption, network data transfer, and balancing service allocation. The number of services allocated is influenced by a variety of factors, including strategy, capabilities, and service requirements. According to the findings, the allocation method and algorithm have a considerable influence on the number of services assigned to the fog and cloud environments. The priority approach allocates more high-priority services to the fog environment and more low-priority services to the cloud environment. The allocation algorithm utilized also influences service allocation, with the worst fit algorithm allocating more services to the fog environment and the first fit algorithm allocating fewer services to the fog environment, but the worst fit can result in more wastage. The results also demonstrate that allocating services without regard for priority or employing a sorting technique without considering priority is inefficient. According to the study, a thorough evaluation of the allocation method and algorithm is required to maximize service allocation in fog and cloud situations.

### 7.2. Resources Usage

The results shown in Figure 5 indicate that the strategy used for distributing services to fog devices can have a considerable influence on the amount of unused RAM in the fog. The best fit method produces the least waste in all circumstances, whereas the worst fit strategy produces the most waste, especially when priority is provided. In terms of waste, the first fit technique lies in between the other two. The chart also demonstrates that high-config devices did not waste any resources because they could handle all of the services assigned to them. These findings imply that a careful study of the allocation approach and algorithm can aid in optimizing the use of fog resources and minimizing waste.

### 7.3. Data Communication over the Network

The results clearly indicate that most of our experiments resulted in low data communication over the network compared to total services allocated to the cloud without strategies Section 6.1 which is the traditional way of allocating services without considering the power of fog and strategies. The variations in fog device capabilities used in our research can help us choose a combination that depends on a number of factors, such as the technical requirements of services as well as the RAM requirements of fog nodes. To decide which allocation technique is optimal for allocating services to fog or cloud devices, we need to consider the service requirements and device capabilities. The results show the amount of time taken to allocate mobile and broadband services to different devices using three different algorithms: worst fit, best fit, and first Fit. The values in the table represent the time taken in hours and minutes for each algorithm and device configuration: low, medium, and high capability. The findings imply that best fit is the most efficient algorithm, as it takes the least amount of time to allocate services for all device types. The worst fit algorithm results in the longest allocation time for low- and medium-capacity devices, while the first fit algorithm falls somewhere in between the other two algorithms. Additionally, the high-capacity devices did not have any allocation time as all services were allocated to them. It is clear that the data communication over the network is reduced with all strategies, and most of the services are allocated to fog devices. Since reducing data transfer across the network was our primary goal, it fits with our proposed strategy. Furthermore, the second aim was to use the priority aspect to allocate the high-priority services as much as possible near the fog devices, and based on the results, this aim was achieved as 90% or more of the services were allocated to the fog devices and the data communication was reduced by 82% compared to Section 6.1.

## 8. Conclusions and Future Work

In conclusion, this paper develops an efficient service allocation strategy priority-based service allocation (PSA) and sort-based service allocation (SSA) with lower bandwidth consumption, faster response times, improved resource usage, and the identification of the best method for processing data at a large scale. Our proposed service allocation strategy was significant because providing services to devices in the IoT is a difficult process due to the many types of devices and their capabilities. Our results showed that by distributing most services locally in fog, we reduced data transmission over the network by 88%, and we maximized the number of distributed services to fog devices by 96%, while minimizing the wastage of fog resources. Our future work includes the following open challenges: Privacy is a concern, as fog nodes acquire a considerable quantity of personal information from fog applications such as smart healthcare. Security is a serious concern, because fog devices lack resources and are positioned in risky environments, which leaves them open to attacks. As a result, designing a lightweight, fast, and reliable security algorithm remains a challenging task.

## Figures and Tables

**Figure 1 sensors-23-07327-f001:**
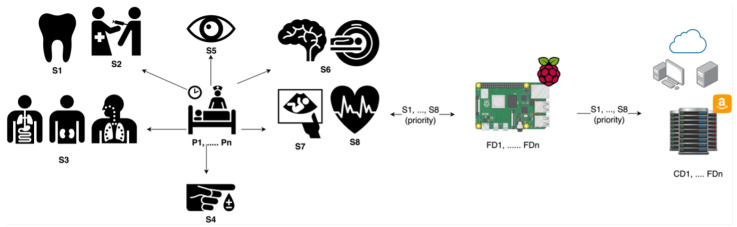
Scenario: a patient in a hospital.

**Figure 2 sensors-23-07327-f002:**
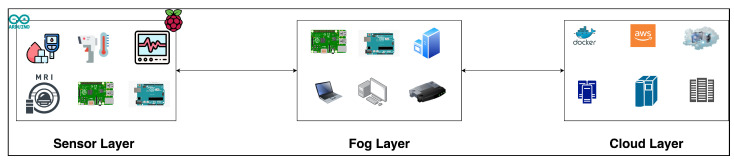
Fog-based IoT architecture.

**Figure 3 sensors-23-07327-f003:**
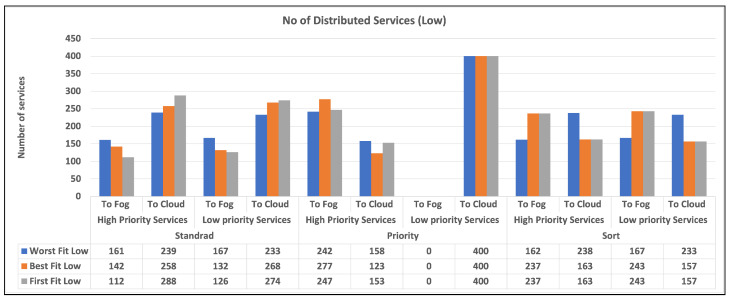
The number of allocated services to fog and cloud (low).

**Figure 4 sensors-23-07327-f004:**
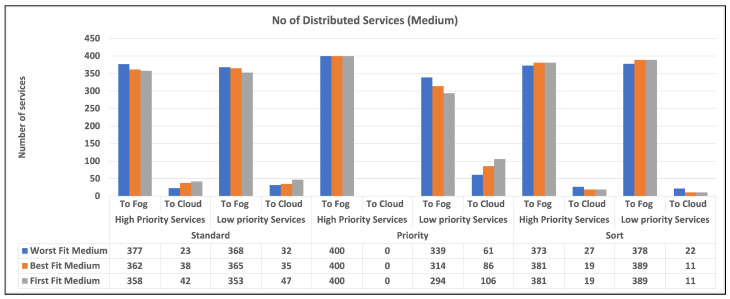
The number of allocated services to fog and cloud (medium).

**Figure 5 sensors-23-07327-f005:**
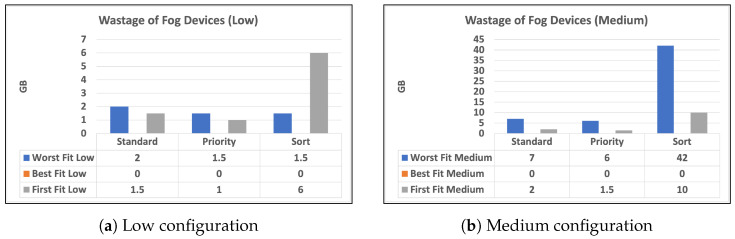
Wastage of fog devices.

**Figure 6 sensors-23-07327-f006:**
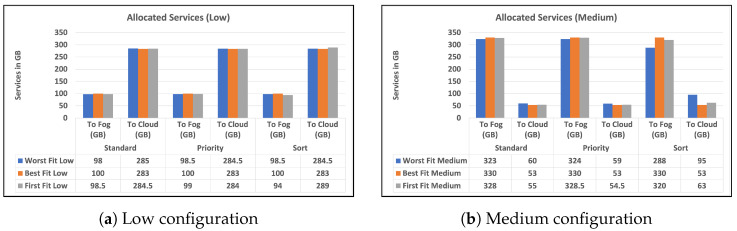
RAM of allocated services.

**Table 1 sensors-23-07327-t001:** Existing works on resource allocation in fog computing.

Authors	Year	Research Focus	Method
Akintoye et al. [38]	2019	Allocating cloud resources	Hungarian algorithm-based binding policy (HABBP) Genetic algorithm-based virtual machine placement (GABVMP)
Abouaomar et al. [44]	2019	Resource allocation	Mobile edge computing application programming interfaces (MEC APIs)
Asensio et al. [39]	2020	Optimal clustering of devices	Fog–cloud clustering (FCC)
Talaat et al. [40]	2022	Resource allocation	Deep-RL, PNN
Talaat et al. [41]	2022	Resource allocation	Optimized RL using PSO
Naha et al. [42]	2022	Energy resource allocation	Multiple linear regression
Dustdar et al. [45]	2022	Management of distributed computing continuum systems	Mathematical artifact called Markov blanket
Li et al. [43]	2023	Resource allocation	Gradient-based algorithms
Hazra et al. [46]	2023	Maintaining energy and delay constraints	Transmission scheduling and computation offloading (TSCO)

**Table 2 sensors-23-07327-t002:** Result of best fit example.

Service Requirement	allocID	Allocated Device
2048	1	FDC4
2048	2	FDC3
2048	3	FDC3
2048	4	FDC2

**Table 3 sensors-23-07327-t003:** The configurations of fog devices in the experiment.

		Fog Devices and Capabilities				
	**FDC Setup**	**FDC1** **2 GB RAM**	**FDC2** **4 GB RAM**	**FDC3** **8 GB RAM**	**FDC4** **16 GB RAM**	**Total FDC (GB)**
1	Low	50 fog Devices	-	-	-	100 GB
2	Medium	15 Fog Devices	15 Fog Devices	10 Fog Devices	10 Fog Devices	330 GB
3	High	-	-	-	50 Fog Devices	800 GB

**Table 4 sensors-23-07327-t004:** The setups of technical requirements of services in the experiments.

	Services and Technical Requirements				
	**SR1** **1 MB–255 MB RAM**	**SR2** **256 MB–511 MB RAM**	**SR3** **512 MB–1 GB RAM**	**SR4** **1 GB–2 GB RAM**	**Total SR (GB)**
No. of services	300 services	300 services	100 services	100 services	383 GB

**Table 5 sensors-23-07327-t005:** No of distributed services low configuration.

	Standard	Priority	Sort
	**To Fog**	**To Fog**	**To Fog**
Worst Fit Low	41%	30.25%	41.125%
Best Fit Low	34.25%	34.625%	60%
First Fit Low	29.75%	30.875%	60%

**Table 6 sensors-23-07327-t006:** No of distributed services medium configuration.

	Standard	Priority	Sort
	**To Fog**	**To Fog**	**To Fog**
Worst Fit Medium	93.125%	92.375%	93.875%
Best Fit Medium	90.875%	89.25%	96.25%
First Fit Medium	88.875%	86.75%	96.25%

**Table 7 sensors-23-07327-t007:** Time of service allocation (fog to the cloud).

	Standard		Priority		Sort	
	**Mobile**	**Broadband**	**Mobile**	**Broadband**	**Mobile**	**Broadband**
Worst Fit Low	3 d 7 h 45 m	23 h 45 m	3 d 7 h	23 h 30 m	3 d 7 h	23 h 30 m
Worst Fit Medium	15 h	4 h 30 m	15 h	4 h 30 m	1 day	7 h 20 m
Worst Fit High	0	0	0	0	0	0
Best Fit Low	3 d 6 h	23 h 25 m	3 d 6 h	23 h 15 m	3 d 7 h	23 h 30 m
Best Fit Medium	12 h	3 h 45 m	12 h	3 h 45 m	15 h 43 m	4 h 40 m
Best Fit High	0	0	0	0	0	0
First Fit Low	3 d 7 h	23 h 30 m	3 d 7 h	23 h 30 m	3 d 8 h	1 d
First Fit Medium	13 h	4 h	13 h	4 h	16 h	4 h 45 m
First Fit High	0	0	0	0	0	0

**Table 8 sensors-23-07327-t008:** Data communication reduction.

	Standard		Priority		Sort	
	**Mobile**	**Broadband**	**Mobile**	**Broadband**	**Mobile**	**Broadband**
Worst Fit Low	25%	26%	26%	27%	26%	27%
Worst Fit Medium	86%	86%	86%	86%	78%	77%
Worst Fit High	100%	100%	100%	100%	100%	100%
Best Fit Low	27%	27%	27%	27%	26%	27%
Best Fit Medium	89%	88%	89%	88%	85%	85%
Best Fit High	100%	100%	100%	100%	100%	100%
First Fit Low	26%	27%	26%	27%	25%	25%
First Fit Medium	88%	88%	88%	88%	85%	85%
First Fit High	100%	100%	100%	100%	100%	100%

**Table 9 sensors-23-07327-t009:** Total services allocation.

Total Services in GB	To Fog	To Cloud	Total
No of Services	0	800	800
Percentage	0	100%	100%
Allocated Services in GB	0	383 GB	383 GB
Time of Allocating Services (Mobile)	0	4 d 11 h	4 d 11 h
Time of Allocating Services (Broadband)	0	1 d 8 h	1 d 8 h

**Table 10 sensors-23-07327-t010:** Execution time.

	Standard	Priority	Sort
Worst Fit Low	155	166	154
Worst Fit Medium	42	53	45
Worst Fit High	37	38	37
Best Fit Low	162	173	99
Best Fit Medium	46	49	41
Best Fit High	37	38	38
First Fit Low	160	166	134
First Fit Medium	47	48	44
First Fit High	37	37	37

## Data Availability

Not applicable.

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
