# Peer review of "Towards an Effective Service Allocation in Fog Computing"

_sensors, 2023, doi:10.3390/s23177327_

Round 1

Reviewer 1 Report (Previous Reviewer 1)

In this article, I still need help comparing related work in other fog computing literature.

Algorithm one and two requires clarification with the help of an example.

 Rest the authors have addressed my comments. 

Moderate English corrections are required. It's better to pass this manuscript from Grammarly or can be proofread by a native speaker one time.

Author Response

Reviewer 2 Report (Previous Reviewer 2)

The major concern of the current manuscript is the lack of clarity in problem context and the problem statement. In addition, the proposed solution and the corresponding setting of services and fog capabilities are too simple to be applied in the real world. For instance, RAM is the only factor considered in service/node matching in the manuscript. In addition, priority (only high or low) is solely used to allocate services to fog or cloud. Several factors (e.g., the processing capability/demand, memory capability/demand, and storage capability/demand) have been discussed in related studies. The novelty, improvement, and contribution of the current manuscript are not clear. The above issues should be addressed first.

Some concerns are listed in the following.

1. It is highly suggested that the problem context, the problem statement, the proposed solution, and the anticipated contribution should be described precisely in the Introduction section.

a. Line 22: The manuscript states that "This suggests that an alternative design is necessary to address these shortcomings." (What are the shortcomings? Are they the focuses of the manuscript?)

b. Line 23: The manuscript states that "Since the IoT architecture connects several devices with varied levels of computing, storage capacity, battery life, and Internet access, device constraint awareness is a crucial part of its design". (Is "device constraint awareness" the focus of the manuscript?)

c. Line 30: The manuscript states that "The main challenge is to decide, while considering overall efficiency, whether services should be run using a fog layer, cloud layer, or a combination of fog and cloud in a certain IoT architecture." (Great point. But it seems that the manuscript tries to allocate all the services in fog.)

d. Line 35: The manuscript states that "Local data storage is an instance when resolving these issues becomes essential." (What are the issues? Is "data storage" the focus of the manuscript?)

e. Line 51: Please define "reaction time". In addition, based on equation (1), the goal is to minimize latency and maximize resource usage. Reaction time is not included.

f. Line 51: Please define "resource usage". 

2. It is highly suggested that the terminologies used in the manuscript should be precise and consistent.

a. Line 33: the term "network edge" is used. Please define and use the term based on the context of the manuscript.

b. Line 39 to 49: Several terms "IoT device", "fog node", "node", "fog device" are shown. Do they indicate the same actor in the context of the manuscript?

3. If IoMT is the focus context of the manuscript, please refer to the research paper "Toward a Heterogeneous Mist, Fog, and Cloud-Based Framework for the Internet of Healthcare Things". Please specify the contribution and improvement of the current manuscript.

4. Please refer to the research paper "Decentralized Edge-to-Cloud Load Balancing: Service Placement for the Internet of Things". Please specify the contribution and improvement of the current manuscript.

5. In line 374. the manuscript states that "If bestFitID remains -1, it indicates that no device is currently allocated to the service, and so the bestFitID is assigned to y." Please check the description with Algorithm 1.

6. A "worst-fit" algorithm is proposed (algorithm 2). Is it possible to use that algorithm in the real world application? Any example? Any benefit?

7. The proposed service allocation algorithm (algorithm 4) is based on priority. However, only low priority and high priority are considered. This might be too simple to be applied in the real world application.

8. Please specify what kinds of services and give corresponding examples in SR1 to SR4. Please describe the design decision of the experiment (e.g., why the numbers and capabilities of FDC1 to FDC4). Thus, readers can get insights into the real world context.

Minor suggestions are listed in the following.

1. Line 361 to line 364 seems to be a duplicate paragraph. 

Please revise and proofread the manuscript.

Author Response

Reviewer 3 Report (New Reviewer)

The authors deal with one of the challenges when working with Fog computing devices, the allocation of services to devices. In concrete, they propose two allocation techniques, Priority-based Service Allocation (PSA) and Sort-based Service Allocation (SSA), with the aim of optimizing resource usage on the Fog and reducing bandwidth utilization. This is achieved by maximizing the number of services allocated in Fog devices, which is the same as minimizing the number of services allocated in Cloud devices. The techniques have been evaluated through use cases related to the healthy domain, where services are related to patients with different priority according to the patients’ status. A dataset composed by both services with different requirements and devices with different capacities, has been used.

From the reviewer’s point of view, the authors tackle a very interesting issue, and the motivational scenario, the experimental setup, and the evaluation are well worked. However, several aspects should be improved as it is stated in the following paragraphs. and in the "Comments on the Quality of English Language" section.

Structure

Section 4.1 describes the PSA and SSA techniques as well as three allocation algorithms. However, the usefulness and necessity of these 3 algorithms is not understood until the techniques are described.

The authors should introduce what and why will be presented in this section, before describing it. This way, the reader could follow the explanation and foresee further issues.

Something similar occurs in Section 7. At the beginning of this section, the authors detail the measures used, which have been calculated in the previous section without specifying which are, and what for they are used.

Overall comments and/or recommendations:

Why several paragraphs are highlighted in yellow?

In the Priority-based Service Allocation, when the allocID is note equal to -1, it is increased by one (line 450). Why? It is not explained in the text.

In general, the writing must be improved. In fact, many sentences are hard to understand or are simply poorly written. The authors must consider rewriting them.

Sentences that are not understood:

* Introduction (lines 21 and 22): “Heavy data transmission via the network is one of the challenges of this design introduced [3].”

* Introduction (line 35): “Local data storage is an instance when resolving these issues becomes essential.”

* Section 4.1 (lines 414 and 415): Following that, we select every service and find out whether it is compatible with the current service.

* May sentences of Section 4.1 used for describing allocation algorithms.

* May sentences and paragraphs employed for explaining obtained metrics must be revised and corrected.

Additionally, the paper is full of spelling errors and typos, which must be resolved. Several examples are listed below:

* Abstract (line 11): “… techniques, which is used to…” => “… techniques, which are used to …”

* Abstract (line 13): minimize => minimizes

* Introduction (line 41): “As the Internet of Things expands, so there is…” => “As the Internet of Things expands, there is …“

* Introduction (line 79): “… Section 6 show…” => “… Section 6 shows …”

* Related work (line 88): “… the fog lead to …” => “… the fog leads to …”

* Related work (line 190): “… describe a resource …” => “… describes a resource …”

* Section 4.1 (line 418): “… the method has three inputs…” => “… the method has two inputs and one output…”

* Section 5.1: lines 525 and 526 should be removed, as they are repeated.

Author Response

Reviewer 4 Report (New Reviewer)

This paper developed an efficient service allocation strategy between priority-based service allocation (PSA) and sort-based service allocation (SSA) with lower bandwidth consumption, faster response times, improved resource usage, and the identification of the best method for processing data at a large scale. This work is novel, and its contributions are well presented. However, the following questions and comments need to be addressed before this paper can be considered for publication.

  1. Time complexity and buffer usage are not studied in this paper. We recommend that the authors conduct a time complexity analysis for the proposed work and compare it with those that are already available. Show the superiority through it. Additionally, the authors should investigate the buffer usage of the proposed work to determine its space efficiency. Finally, the authors should analyze the scalability of the proposed work, to assess its potential for use in larger systems.

  2. From the objective function, What are the range of values considered for w1 and w2? is it like w1+w2 =1?

  3. from equation 1, f_m <- instead of fm

  4. Where does eq1 used in the algorithms?

  5. From Table 1, only six works are studied, eventhrough there are seveal aricles in the literature. Is there any specific reason for this? It is also noted that, the limitations of the existing works are not mentioned in this table. Recommended to add another column to this table and discuss the limitations. Recommended to look in to most recent works such as 'Cooperative transmission scheduling and computation offloading with collaboration of fog and cloud for industrial iot applications', 'On distributed computing continuum systems'.

  6. Why does the authors do not mention the equation number in section 4? Where does these equations used in the proposed algorithms?

  7. Section 5.1 is unclear. The authors are recommended to provide clear details including citations for the datasets. If the data generated synthetically, explain the ways the authors generated it. If you consider from any realtime data through hospitals, clearly mentioned it. Provide specific information on the methods used to collect the data, and any preprocessing steps that have been taken. Describe the data set in detail, including the type of data (images, texts, etc.), the quantity of data, and the source of the data. Include a link to the data set if possible.

  8. Protocols are very improtant in data exchanging between the devices, but the setup does not talk about any protocols. Will the authors able to specify which protocols used during experiments? Such as 'Survey on recent advances in IoT application layer protocols and machine learning scope for research directions'. This will help to improve the understanding of the process and enable further research in this field. Additionally, it will provide valuable insight into the effectiveness of the protocols used.

  9. Summarize the limitations of proposed work. It is also recommended to provide the reasons for achieving the superior performance of the proposed work over existing ones. However, the proposed work still has some limitations, such as the restricted range of data it can process. Exploring the reasons for superior performance can help to improve the proposed work and make it more effective in the future.

Round 2

Reviewer 2 Report (Previous Reviewer 2)

The major issues are address in the manuscript. The effort of the authors are highly appreciated.

It is suggested that the authors can proofread the manuscript again because lots of modifications are made.

It is suggested that the authors can proofread the manuscript again because lots of modifications are made.

Author Response

Reviewer 4 Report (New Reviewer)

No further comments to suggested

Author Response

This manuscript is a resubmission of an earlier submission. The following is a list of the peer review reports and author responses from that submission.

Round 1

Reviewer 1 Report

This paper presents an efficient allocation technique that pushes processing closer to the network’s fog side. It vestigates which devices and services may best allocate while preserving resource usage in the IoT architecture.

The paper is well-written and well organised

Please add the comparison in related work other fog computing literature.

Scenarios need to be explained further also assumptions need to be discussed in a sentence than bullet points.

Explain algorithms 1 and 2 with examples, please

Please merge the conclusion and future work together in a single section and without bullet points

English styles requires minor corrections

Reviewer 2 Report

The manuscript proposes Priority-based Service Allocation and Sort-based Service Allocation algorithms to deploy services to fog/cloud environments effectively. The deployment goal of the proposed algorithms is to maximize the resource usage of fog environments and then minimize the data communication. Algorithms are introduced and the corresponding experiments are designed and performed to present the performance of the proposed algorithms.

Major concerns of the manuscript includes 1) the research goal and the state quo of related works 2) the design of the experiment and 3) the application domain.

It is highly suggested that the manuscript can be revised and the readers of the journal can get more insights into the contribution of the manuscript.

1) the research goal and the state quo of the related works.

a. The research goal of the proposed manuscript is to achieve efficient resource allocation among fog/cloud environments. The attributes considered in the algorithms include the capability of fog devices (i.e., memory) and the resource requirement (i.e., memory) of services. It might be too simple to use in the real world (or medical domain). For instance, the computing power, response time requirement, execution time requirement, the service priority, and so on, can be considered and modeled in the algorithms. Thus, the proposed algorithms can show their effectiveness and be deployed in the real world problems.

b. The issues of resource allocation among fog/cloud environments are addressed in many studies. It is suggested that the authors can elaborate on the remaining challenges of the previous studies, and the specific challenge discussed in the manuscript. Thus, readers can get more insights into the research question, the proposed solution, and the contribution of the manuscript.

2) the design of the experiment

a. Some information about the configurations of fog devices and the characteristics of services are presented. Is there any base for the configuration? Does it come from a real world environment (i.e., hospital) or real scenario? 

b. It is suggested that the authors can elaborate on the method of the experiment. Is any simulator used in the experiment? How the results are generated?

c. In the experiment results, the service allocation is performed through mobile network and broadband. It might not be a good idea to connect fog and cloud environment by mobile network in the medical domain. In addition, the comparison between the performance brought by mobile network and broadband does not present the benefit of the proposed algorithms.

d. In the experiment results, the performance of high level configuration of fog devices does not show because all the services are handled in the fog. If it is the case, why design this experiment setting? What is the base?

e. It is highly suggested that several bases from related studies can be used for the comparison and evaluation of the proposed algorithms. For instance, the proposed algorithm achieves certain improvement when comparing to other allocation algorithms mentioned in previous studies. Otherwise, readers cannot realize the contribution and the benefit of the manuscript.

f. The configuration of service priority is too simple.

3) the application domain: In the manuscript, IoMT (Internet of Medical Things), hospital, and health care system are mentioned. In the context of medical service, the necessary resources might be allocated firmly and exclusively. For instance, data from a specific set of sensors will be collected and analyzed by associated services. This might not change at runtime. If it is the case in the medical domain, it is suggested that the authors can introduce the actual case or scenario for readers.

Round 2

Reviewer 1 Report

The authors have addressed most of my comments except  conclusions

The conclusions are too big. It can be reduced to 8 to 10 lines focusing on core contribution. In addition, please rename the section as "Conclusions and Future Work"

The quality of English is good, and slight proofreading is required

Reviewer 2 Report

Several issues and comments are addressed in the current manuscript. The efforts of the authors are appreciated.

However, the contribution of the manuscript is hindered by the proposed modeling.

1. It is highly suggested that the real hospital case can be introduced. The actual IoMT, specific fog nodes, and cloud services in the case can be described. This helps readers to understand the context and the problem comprehensively. 

2. Based on current manuscript and the response to review comments, the computing power, response time requirement, and execution time requirement are not considered in the proposed service allocation model because IoMT has limited resources. However, the proposed model aims to handle (all) services in the fog layer. If fog is such a major role in the specific application environment, memory requirement and capacity should not be the only consideration. 

3. Based on related works mentioned in the manuscript, several studies consider capacity, RAM, CPU, and so on in the model. The reason why memory is considered only in the current manuscript should be explained. In addition, the advantage or improvement should be evaluated when comparing with previous studies.